# The Impact of *OsERF34* on Rice Grain-Processing Traits and Appearance Quality

**DOI:** 10.3390/plants14111633

**Published:** 2025-05-27

**Authors:** Zhimin Du, Yinan Jia, Peisong Hu, Hai Xu, Guiai Jiao, Shaoqing Tang

**Affiliations:** 1Rice Research Institute, College of Agronomy, Shenyang Agricultural University, Shenyang 110866, China; 2021200060@stu.syau.edu.cn (Z.D.); jyn0227@hotmail.com (Y.J.); 2China National Rice Research Institute, Hangzhou 311401, China; hupeisong@126.com

**Keywords:** AP2/ERF transcription factor, appearance quality, processing quality, physiochemical properties, head rice rate

## Abstract

The head rice rate, defined as the proportion of milled grains retaining at least three-quarters of their original length, has become a limiting factor that restricts the improvement of rice quality in China in recent years. Here, we characterized the role of ETHYLENE RESPONSIVE FACTOR34 (OsERF34), an APETALA2 (AP2/ERF) family TF, in the grain morphology, physiochemical properties, and processing quality of rice. Through CRISPR/Cas9-mediated knockout (*Oserf34*) and overexpression (OsERF34-OE) in the japonica cultivar ZH11, we demonstrate that *OsERF34* exerts dose-dependent effects on grain morphology and processing traits. *Oserf34* mutants exhibited significantly elevated chalkiness levels, with a 52.0% increase in percentage of grains with chalkiness(PGWC) and a 65.4% enhancement in chalkiness degree, with disordered and enlarged starch granules, reduced amylose content and skewed chain-length distribution (A/B1 chains increased but B2/B3 chains decreased), and displayed heightened starch solubility and swelling power but diminished milling resistance (shear hardness having fallen by 12.7–16.1% and compression hardness having fallen by 11.2–16.4%), culminating in doubled breakage rates and lower head rice rate (decreased by 6.7–9.0%) during processing. Strikingly, both mutants and OE lines showed analogous grain narrowing, yet the processing quality diverged. Mutants suffered structural fragility, while the OE lines enhanced mechanical robustness (compression hardness increased by 11.4–12.1%). The OsERF34-OE lines achieved 6.5–7.1% higher head rice rates. Our work positions *OsERF34* as a dual-function regulator that governs grain morphology, regulating appearance and processing quality. These insights suggest that an overexpression of *OsERF34* could improve processing efficiency, potentially laying a foundation for precision breeding.

## 1. Introduction

Rice (Oryza sativa) is a staple food for more than half of the global population [1]. It is mainly consumed as cooked grains, and its quality is crucial for consumer preference, processing efficiency, and market value. Grain appearance, milling, sensory evaluation, and cooking are the four main aspects that determine grain quality, promising consumer acceptability [2]. Crystal clear with low chalkiness of rice grain is in line with consumers’ preference. Good processing characteristics with a high head rice rate meet the needs of rice processing enterprises. Now, there are six grading indexes, including the head rice rate, chalkiness degree, transparency, amylose content, gel consistency, and alkali spreading value, in the current cooking rice variety quality standards, according to cooking rice variety quality. The agricultural industry standard of the People’s Republic of China, NY/T 593-2021 [3,4], targeted the continuous improvement of rice varieties, with the rates for reaching the standards for amylose content, gel consistency, and alkali spreading value achieving quite a high level, some even more than 90%. However, the head rice rates of different types of rice varieties were generally low, as shown in Appendix A. The success rate of head rice of the regional test japonica variety samples was only 58.97% and 45.31% in the year of 2023 and 2024, much lower than other indicators [3,4]. This suggests that head rice rates have become a limiting factor that restricts the improvement of rice quality in China in recent years. This systemic deficiency of low head rice rate underscores the urgent need for targeted research integrating genomic insights with precision agronomic practices to enhance the head rice rate in modern rice production systems.

According to the national standard “Milling and Breaking Degree of Rice” (GB/T 21719-2008) [5], head rice is defined as the proportion of milled grains that retain at least three-quarters of their original length. The head rice rate is the percentage of head rice relative to the mass of the cleaned rice sample [5]. Previous studies have predominantly focused on the impacts of environmental factors such as sowing date, fertilization regime, harvest and storage practices, as well as temperature and light conditions during the growth and postharvest stages on processing quality [6,7,8,9,10]. Numerous studies have confirmed the interlinking effects between different rice quality traits, including increased grain length. The length-to-width or length-to-thickness ratios negatively regulate rice milling quality, whereas increased grain width and thickness could improve rice milling quality [11]. High-chalkiness varieties generally exhibit a lower head rice rate, indicating that visual selection based on PGWC (percentage of grains with chalkiness) may serve as a rapid and simple screening method for the head rice rate [12]. Although several QTL loci that regulate processing quality have been documented in prior studies [13,14,15,16,17,18,19], the molecular mechanisms underlying rice milling quality formation remain poorly elucidated, and the genes directly associated with these traits are largely uncharacterized. As a complex agronomic trait, the molecular genetic regulation mechanism of the head rice rate is still relatively unclear. Only a limited number of genes that regulate the head rice rate have been functionally characterized. The loss function of *Chalk5* results in disturbing the pH homeostasis of the endomembrane trafficking system in grains, which in turn, affects the biogenesis of protein bodies and forms more small vesicle-like structures, causing the formation of air spaces in the endosperm, indicating that the chalkiness probably affects the head rice rate [20]. Given this context, in addition to considering the influencing factors for head rice rate, such as planting conditions, environmental conditions, storage, and processing procedure, enhancing the head rice rate of a variety itself is still the most critical. Therefore, studying the complex genetic and molecular mechanisms affecting head rice is particularly important for rice quality improvement.

APETALA2/ethylene response factors (AP2/ERFs) are widely known plant-specific TFs that play crucial roles in the transcriptional regulation of plant growth and development [21,22,23]. There are 139 AP2/ERF family genes in rice [24]. Many AP2/ERF family genes were identified in the plant genome. They can be divided into five main groups based on the number of AP2/ERF structural domains, namely AP2 (APETALA2), ERF (ethylene-responsive factor), DREB (dehydration-responsive element binding protein), Soloist, and RAV (related to ABI3/VP1) [25]. In addition to the significant regulatory roles of AP2/ERFs in abiotic stress responses [26], recently, several studies have implicated their roles in plant development and crop improvement. A study indicated the novel regulatory roles of AP2/ERFs in fruit ripening or secondary metabolite production in plants [27,28,29]. *OsERF34* was identified as directly promoting the expression of the rice morphology determinant (RMD) in rice peduncles, positively regulating secondary cell wall synthesis and mechanical strength in rice [30]. Above all, the concerns of the role of AP2/ERFs on rice quality are still far from sufficient. Their functions need to be explored deeply and comprehensively from different angles.

In our study, the qRT-PCR analysis revealed pronounced transcriptional activity of *OsERF34* specifically localized in developing the rice endosperm (Figure 1E). This spatiotemporal expression pattern strongly suggests its regulatory predominance in endosperm morphogenesis, potentially orchestrating starch deposition patterns that ultimately govern grain quality determinants such as translucency and milling characteristics. To elucidate the genetic regulation of grain quality, we generated *OsERF34* loss-of-function mutants (*Oserf34*) and overexpression (OsERF34-OE) lines in the ZH11 genetic background. A comprehensive comparative analysis was conducted, including a phenotypic characterization of grain morphology, starch granule architecture, X-ray diffraction profiling of crystalline starch structures, biochemical quantification of endosperm composition, thermodynamic assessment of starch physicochemical properties, and industrial evaluation of milling performance were investigated in this study, aimed at revealing the *OsERF34*-dependent modulation on rice quality and providing an actionable strategy to enhance the head rice yield while improving market-grade rice proportions.

## 2. Results

### 2.1. Structure and Functional Importance of the AP2 Domain

The APETALA2/ethylene-responsive element-binding factor (AP2/ERF) is a plant-specific TF containing one or two DNA-binding domains and modulates the expression of downstream genes to regulate plant tolerance to environmental stresses. First, we find out that the AP2 domain, which is conserved in the gene structure of AP2/ERF TF (Figure 1A), is involved in key roles like growth, differentiation, and responses to environmental stresses. We have checked the phylogeny and multiple amino acid sequence alignment of OsERF34 in both indica and japonica rice. The sequence alignment revealed complete sequence similarity (100%) in the coding region, demonstrating that the OsERF34 protein exhibits a highly conserved nature across these two cultivated rice varieties. Furthermore, OsERF34 maintains 95% amino acid homology with homologs in Brachypodium distachyon and Triticum turgidum subsp. durum, which indicates that *OsERF34* likely encodes a functionally critical protein that is essential for the survival of Poaceae plants. (Figure 1B,C).

We also checked the expression patterns of *OsERF34* in different tissues of rice and the developing seeds of rice and compared those with the eFP browser (https://www.bar.utoronto.ca/efp//cgi-bin/efpWeb.cgi, accessed on 22 February 2025, Figure 1D). Our qRT-PCR analysis showed that the expression of *OsERF34* in plant morphological structures was comparatively lower. In the roots and shoots, the expression of *OsERF34* was lower than in the leaves and internodes. However, its expression was much higher in the endosperm, indicating its critical role in rice endosperm development and the formation of rice grain quality (Figure 1D,E). Based on these findings, we designed a CRISPR/Cas9-mediated knockout strategy targeting *OsERF34* (Os04g0550200) to validate its functional role. The resulting loss-of-function mutant *Oserf34* was successfully generated, with chromatograms documenting the mutagenesis events in both the wild-type ZH11 and mutant sequences provided in Appendix A.

### 2.2. OsERF34 and Its Impact on Rice Morphology

The schematic diagram shows the *OsERF34* gene with the CRISPR-Cas9 target site (Figure 2A). We also checked the expression patterns of the gene mutated and compared with the WT ZH11 and found its lower or completely diminished expression in the leaves of rice, confirming the complete knockdown of *OsERF34* in our experiment (Figure 2B).

When we find out from the tissue expression that the *OsERF34* is strongly expressed in the endosperm of rice seeds, we wonder about the morphology of the seed. Interestingly, we observed significant differences in the lengths of the rice grains (Figure 2D,E). However, the grain width decreased significantly in *Oserf34* compared to WT, indicating its role in regulating grain size, which possibly influences cell proliferation (Figure 2D,F). Further investigating the length–width ratio suggested an increased ratio in the mutants compared to WT, which alternatively decreased the thousand-grain weight in the mutants (Figure 2H,C). Similarly, a reduction in grain thickness in the mutants was also noticed (Figure 2G,D). Our investigation suggests that *OsERF34* might possibly be involved in the genetic pathways that control grain morphology, and its downregulation could lead to narrower grains.

### 2.3. Comparison of Chalkiness Phenotype and Starch Granules Morphology Between ZH11 and Oserf34

We compared the chalkiness property of the WT and mutants. The whole grain appearance indicated that the grains were more uniform with minimal chalkiness in the WT (Figure 3A); the mutants showed highly chalky grains with more porous structures (Figure 3C). Similarly, SEM micrographs of the rice endosperm at high magnifications also showed looser and irregular starch granules (Figure 3B–D). We also observed a significantly high PGWC (67.8–68.2%) and chalkiness degree (38.5–65.4%) for the mutants compared to the WT (Figure 3F,G). Furthermore, the granulometric distribution and diameter of the starch granules also indicated a larger size with less extrusion in the mutants compared to the WT (Figure 3E,H,I). All above, we interpret that *OsERF34* plays an important role in regulating starch granule morphology and affects the appearance quality ultimately.

### 2.4. Comparison of Physiochemical Properties Between ZH11 and Oserf34

Physiochemical properties of the rice endosperm are also of prime importance while checking grain quality traits. Therefore, we measured the total starch content and fat content and observed non-significant differences between the mutants and WT (Figure 4A,C). However, highly significant differences were observed for protein content and soluble sugar between the mutants and WT, yielding an increased level of protein and sugar contents in the mutants (Figure 4B,D). Amylose content, a key determinant of rice cooking and appearance quality, exhibited a significantly greater reduction in the mutants compared to ZH11 (Figure 4E). Similarly, gel consistency, an established indicator of starch colloidal properties that correlates with the hardness of cooked rice texture, showed marked decreases in *Oserf34* (Figure 4F). Due to the loose starch granule composition, we speculate that water and heat are more easily able to penetrate into the gaps between rice starch granules, leading to higher solubility and swelling power. As expected, we observed higher solubility and swelling power in the mutants (Figure 4G,H).

We also measured the viscosity characteristics and pasting properties. The viscosity profiles of *Oserf34* demonstrated a reduction in peak viscosity relative to the wild-type ZH11, with concomitant and significant decreases in trough viscosity and final viscosity (Figure 4K). The gelatinization characteristics of endospermic starch were comprehensively evaluated using differential scanning calorimetry (DSC). The analysis revealed that both the peak temperature (Tp) and conclusion temperature (Tc) for *Oserf34* were significantly lower than those of ZH11. Conversely, the gelatinization enthalpy (ΔH) of endospermic starch in the *Oserf34* mutants exhibited a notable increase (Figure 4J). This may be largely due to its loose starch granules structure with more space for expansion, which leads to more heat energy being required to break the starch granules down into a paste.

The architecture of amylopectin branches and chain arrangements directly influences the crystalline structure and, consequently, the functional properties of starch [31,32,33,34]. Here, we also measured the starch’s fine structure and the starch granules’ crystallinity. Notably, the differences in the starch chain’s length distribution were observed between ZH11 and *Oserf34*. Compared to the wild-type ZH11, the *Oserf34* mutants exhibited a significant increase in A-chains (DP 6–12) and the majority of B1-chains (DP 13–24). Conversely, B2-chains (DP 25–36) and B3-chains (DP > 37) showed a marked reduction (Figure 4L,M). Accordingly, a highly significant difference in relative crystallinity was observed between the wild-type ZH11 and *Oserf34* mutants, with the mutants exhibiting markedly higher values (Figure 4I). These differences were also important reasons for the variances in their cooking properties

### 2.5. Processing Quality and Grain Hardness Characters of ZH11 and Oserf34

The mutants showed a highly significant reduction in the head rice yield rate (6.7–9.0%) compared to ZH11 (Figure 5A). Rice grain breakage primarily results from excessive bending and tensile stresses induced by compressive and shear forces during processing. In this study, we employed compression and shearing modes to simulate the operational principles of two distinct huller types utilized in rice processing, namely roller extrusion (corresponding to compression) and rotary impact (corresponding to shear). Comparative analysis revealed that *Oserf34* mutants exhibited significantly reduced shear hardness (27.6–34.9%), shear energy (33.9–38.5%), and shear fracture force (36.3–46.5%) relative to wild-type ZH11 under shear loading (Figure 5B–D). Compression testing demonstrated that *Oserf34* grains displayed diminished mechanical robustness, with lower compression hardness (16.4–24%), reduced fracture force (26.7–29.0%), and decreased energy absorption (34.0–37.6%) compared to WT (Figure 5E–G), indicating enhanced structural fracture and increased susceptibility to compression-induced damage. This structural fragility aligns with the mutants’ heightened vulnerability to milling-induced fractures. No significant differences were observed in the brown rice rate (Appendix A) or milled rice recovery (Appendix A) between ZH11 and *Oserf34*. Collectively, these findings demonstrate that *OsERF34* critically regulates grain quality traits by modulating the mechanical integrity of the rice endosperm. The compromised cellular architecture in the *Oserf34* mutants directly correlates with processing-related breakage, providing mechanistic insights into genotype-specific milling resistance.

### 2.6. qRT-PCR Analysis of Starch Synthesis and Grain Morphology Related Genes

The findings collectively reveal that *OsERF34* modulates the physicochemical properties of endosperm starch through a precise regulation of amylopectin chain length distribution and relative crystallinity. The loss-of-function mutation induces structural loosening of the starch architecture, ultimately causing significant deterioration in rice processing quality. Based on this, we also carried out a qRT-PCR of the genes related to starch synthesis in *Oserf34* and ZH11 (Figure 6). Our analysis showed that the function deletion of the *OsERF34* gene led to changes in the expression of many genes related to starch synthesis, in addition to *OsBEI* and *OsPUL3*, three glucose pyrophosphorylase genes (*OsAGPL2*, *OsAGPL3*, *and OsAGPS2b*) responsible for substrate provision in starch biosynthesis that exhibited significantly downregulated expression in the mutant. As these genes encode key rate-limiting enzymes in starch synthesis, their reduced expression may lead to insufficient substrate availability for starch biosynthesis. This substrate limitation likely triggers a physiological feedback regulation mechanism, resulting in significant upregulation of other starch synthase components. These observations suggest that the glucose pyrophosphorylase genes may serve as potential targets of OsERF34 in starch biosynthesis regulation.

### 2.7. Comparative Analysis of Key Phenotypes Between ZH11 and OsERF34-OE

In terms of physicochemical properties, the overexpression lines exhibited a significant increase in total starch content and amylose content compared to the wild type (Figure 7A,B). However, no significant differences were observed in protein content, fat content, soluble sugar content, and gel consistency between the overexpression lines and the wild type (Appendix A). Furthermore, the water solubility showed minimal differences between the OsERF34-OE lines and the wild-type ZH11 rice grains (Appendix A). Intriguingly, compared with ZH11, the overexpression lines exhibited a significant reduction in swelling capacity, which is precisely contrary to the phenotypic changes observed in the knockout mutants (Appendix A). The viscosity profile of starch in the overexpression lines closely mirrored that of ZH11, with the peak viscosity and trough viscosity values remaining consistent with the wild type (Appendix A). The gelatinization properties of endosperm starch were also evaluated. Differential scanning calorimetry (DSC) analysis revealed that the peak temperature (Tp), conclusion temperature (Tc), and gelatinization enthalpy (ΔH) were all significantly reduced in the overexpression lines compared to ZH11 (Appendix A). Additionally, the starch chain length distribution in the overexpression lines exhibited an inverse pattern relative to the wild type, characterized by a reduction in short chains, an increase in medium and long chains, and a decrease in relative crystallinity (Figure 7C,E).

However, the head rice yield increased significantly by 6.5% and 7.1%, respectively (Figure 7D). Subsequent testing of hardness traits in the overexpression lines demonstrated significantly elevated compression hardness and compression work (Figure 7F,G), while the shear properties showed no notable differences (Appendix A). The length–width ratio of the grains in the overexpression lines was significantly greater than that of the wild type ZH11 (Appendix A). Further comparative analysis of the chalkiness characteristics between the wild type (ZH11) and OsERF34-OE revealed that both exhibited uniformly shaped grains with minimal chalkiness and tightly packed starch granules (Appendix A). A comparative analysis of starch granule size distribution and structural morphology between OsERF34-OE lines and ZH11 grains revealed that one overexpression line exhibited significantly reduced granule dimensions relative to the wild type, whereas the other line showed no statistically discernible variation (Appendix A). We also observed no significant difference for thousand-grain weight, brown rice and milled rice rate, and grain transparency between the OE lines and WT (Appendix A).

## 3. Discussion

Rice grain quality is an important agronomic trait and has been in the spotlight for a long time. To date, several genes and pathways controlling grain quality, composition, and size in rice have been studied through conventional map-based cloning and reverse genetics techniques [35,36,37,38,39]. The AP2/ERF family is identified as involved in various aspects of plant development, drought, and submergence stress regulation processes; fruit ripening or secondary metabolites production; as regulators of cell wall biosynthesis; and so on [40,41,42]. There are very few studies on the effects of the AP2/ERF family on rice grain formation and quality. OsERF115 targets the GCC box of the *OsNF-YB1* promoter to regulate endosperm development and grain filling in rice. *OsNF-YB1* is specific to the aleurone layer of developing endosperms [43]. According to our observation, *OsERF34* exhibited lower expression in vegetative structures but was highly expressed in the endosperm, suggesting its crucial role in regulating stored substances in developing endosperm. Here, we discovered a new function of *OsERF34* that is directly involved in rice grain quality, which can reduce the chalkiness degree significantly and enhance the processing characteristics and processing quality greatly. The targeted mutation site in the mutant used to validate the function of this gene is located in the mid-to-late region of the gene. Importantly, these mutations occur within the functional AP2 domain and induce a frameshift mutation that alters all downstream amino acids. We have experimentally verified the three-dimensional protein structure and confirmed that the mutation completely disrupts the native conformation. This structural perturbation strongly supports the functional null nature of the mutant protein.

There are still relatively few studies on the genes that affect the rice processing characters and processing quality. *FED1*, an important QTL site that determines rice crack resistance and head rice rate, was revealed by genome-wide association analysis. *FED1* is *Waxy*/*Wx*, the coding gene of granular binding starch synthase I. The alleles of the *Wx* gene, *wx*, *Wxb*, *Wxa*, and *Wxin*, give different crack resistance to rice. Among them, the rice with the ineffective mutation wx has the highest crack resistance, while the rice with a high *Wx* expression of *Wxa* and *Wxin* significantly reduces the crack resistance [44]. Additionally, studies have reported that *Wx* and *ALK* coordinately regulate the head rice rate by affecting amylose content, the number of amylopectin branches, amyloplast size, and thus, grain filling and hardness [45]. On the whole, we need to further explore more genes that regulate rice processing characteristics and elaborate on their influence mechanisms on processing quality.

Grain quality in rice is often determined by a combination of properties, like chemical, physical, and sensory properties. Collectively, these traits influence rice appearance, nutritional value, processing efficiency, and consumer preferences, with grain morphology and head rice rate being critical for market value.

Physical traits like grain length, grain width, and thickness are essential for grain yield and machine design used in handling, separating, drying, and storage. The length-width ratio of the grain determines the shape of the grain, while the grain length shows the size of the grain. We observed an increase in the grain length and length–width ratio of the *Oserf34* mutants, which indicated an influence of the *OsERF34* on the rice grain morphology. However, grain width and grain roundness decreased significantly, which alternatively reduced the thousand-grain weight in the mutants. Similar observations were recorded for the knockdown of *RISBZI*, which reduced the seed size and starch content [46].

Amylose content is determined as the principal influencing sensory entity of rice consumption, which directly correlates with rice grain qualities such as color, tenderness, cohesiveness, and glossiness [47]. However, environmental factors, particularly ambient temperatures, may regulate the amylose levels during the crop-ripening stages, as evident from the literature [47,48,49,50]. Research has found that amylose content is correlated with many rice quality traits [47,50]. Consistently, *Oserf34* mutants exhibited significantly lower amylose content and gel consistency and higher protein and soluble sugar contents, whereas the OsERF34-OE lines showed significantly higher amylose content and total starch content, with no notable difference in other physicochemical parameters compared to WT ZH11. Conclusively, variations in amylose content among rice varieties highlight the complexity of starch composition regulation, which ultimately influences rice texture and quality. Therefore, we proceeded to further investigate the starch-related traits.

In the investigation of starch fine structures, we found that the starch granules in *Oserf34* mutants became larger, irregular in shape, loosely packed, and exhibited increased relative crystallinity. Further analysis of starch chain length distribution revealed an increase in short- and medium-length chains and a decrease in long chains. In contrast, the OsERF34-OE lines showed the opposite trends in starch fine structures. This result indicates that starch synthesis and distribution are affected during the grain-filling process. To date, most of the starch biosynthesis-related genes have been studied, such as soluble starch synthases (*SSSs*), granule-bound starch synthases (*GBSSs*), debranching enzymes (*DBEs*), ADP–glucose pyrophosphorylases (*AGPases*), starch-branching enzymes (*SBEs*), and starch phosphorylase (*PHO*) [51]. According to our qRT-PCR analysis of genes related to starch synthesis, three glucose pyrophosphorylase genes (*AGPLs*) responsible for substrate provision in starch biosynthesis exhibited significantly downregulated expression in the mutant. This substrate limitation likely triggers a physiological feedback regulation mechanism, resulting in significant upregulation of other starch synthase components (*OsSSIIa*, *OsSSIIIb*, *OsPHO1*, *OsPHOH*, *OsPK1*, *OsPK2*, et al.). These observations collectively suggest that the glucose pyrophosphorylase genes may serve as potential targets of OsERF34 in starch biosynthesis regulation. However, this hypothesis requires further experimental validation through subsequent molecular investigations. It is well-established that starch biosynthesis constitutes a complex biological process regulated by multiple genes. An alteration in a single gene resulting in the loss of its function may trigger the upregulation of numerous functionally related genes to compensate for the disrupted metabolic processes and maintain normal physiological homeostasis. Our quantitative analysis revealed that the majority of starch biosynthesis-related genes exhibited upregulated expression patterns, while a minor subset showed downregulation. We hypothesize that these downregulated genes are potential downstream targets of OsERF34, although this speculation requires further experimental validation. In our study, *OsERF34* has been demonstrated to participate in modulating both starch synthesis and catabolic pathways, thereby influencing starch crystalline structure and chalkiness characteristics, which ultimately leads to modifications in head rice yield through these coordinated regulatory mechanisms.

Chalkiness, which generally refers to the opaque areas in the sperm, is another factor that affects the milling property of rice grains. The loose endosperm structure in chalky grains exhibits reduced mechanical strength, making them more susceptible to breakage during milling processes and consequently lowering head rice yield. We observed a significantly higher percentage of partially filled grains PGWC (52.0%) and chalkiness degree (38.5–65.3%) in the mutants compared to the WT. These findings suggest that *OsERF34* plays a crucial role in regulating grain translucency and endosperm structure, ultimately affecting rice quality. The increased chalkiness in the mutants may be attributed to alterations in starch accumulation, granule organization, or protein matrix formation. Since chalkiness negatively impacts milling efficiency and market value, these results indicate that *OsERF34* significantly influences rice grain appearance, processing quality, and overall consumer preference.

In addition to rice grain shape, starch granule synthesis and composition are other milling properties that impact grain strength, affecting the chalkiness and breakage properties of rice grains. A study observed decreased starch accumulation in *RPBF* knockdown grains due to a lower *GBSSI* differential expression in the mature grains [52]. Our investigation showed non-significant differences for starch accumulation between WT and *Oserf34*. However, the values of starch contents were high in the OsERF34-OE lines, indicating their potential role in regulating starch biosynthesis and deposition in rice grains. The arrangement of starch granules in rice grains affects the grain shape, chalkiness, and transparency. For instance, a looser arrangement of starch granules in the grains causes cavities between the starch granules, which alternatively reduce transparency in rice grains, resulting in an opaque or chalky endosperm [53]. This phenomenon was evident in our observation. The milling property is crucial for maximizing rice yield and refers to the dehulling and polishing of rice grains. Our investigation showed that the overexpression lines exhibited increased grain hardness, tightly packed and regularly arranged starch granules, and a significantly improved head rice recovery rate, which contrasted sharply with the grain appearance and processing quality of the *Oserf34* mutants. Although the grain shape of the overexpression lines became longer and narrower, this change mirrored the grain morphology observed in the mutants. Combined with the above analysis results, *OsERF34* regulates the head rice rate by affecting amylose content, the number of amylopectin branches, amyloplast size, and thus, grain filling and hardness, rather than altering the grain morphology. This is consistent with the function of gene *OsDOF18* reported in previous studies [45]. Furthermore, a substantial body of research has demonstrated the interconnected effects of starch, chalkiness, and head rice rate in the endosperm. Moderate soil drying (MD) has been demonstrated to stabilize starch’s molecular structure and functional properties under high-temperature conditions, thereby enhancing the head rice rate while reducing grain chalkiness [54]. Furthermore, *OsATL13* principally transported phenylalanine and methionine. The upregulation of *OsATL13* modulates the expression of genes associated with starch metabolic pathways, and exogenous application of phenylalanine and methionine significantly improves the head rice rate and decreases the chalky grain incidence [55]. Moreover, *qMq-1* and *qMq-2*, predicted as allelic variants of *Wx* and *ALK*, coordinately regulate rice milling quality through changing the starch physicochemical properties and hardness of the kernel, not by grain shape or chalkiness, which were confirmed by transgenic complementary and overexpression tests [45]. Collectively, these findings corroborate the mechanistic linkages between starch architecture, chalkiness traits, and head rice rate optimization established in the present study.

## 4. Materials and Methods

### 4.1. Plant Materials and Growth Conditions

The rice (*Oryza sativa* L.) genotypes used in the present study were Zhonghua11 cultivars. All the plants were grown under natural field conditions at the China National Rice Research Institute, Hangzhou, China (30°15′ N). Rice seeds are typically sown around May 25 and transplanted around June 15, with a planting density of 25 cm (row spacing) × 10 cm (hill spacing). Field management follows standard paddy field management practices. All experimental materials were planted simultaneously in the same paddy field under uniform water and fertilizer management protocols to ensure consistency across trials. Chemical fertilizers were applied at the following rates: urea: 465 kg/ha; diammonium phosphate (DAP): 150 kg/ha; and potassium chloride (KCl): 187.5 kg/ha. This corresponds to elemental nutrient applications of nitrogen (N): 240 kg/ha; phosphorus (P): 69 kg/ha; and potassium (K): 112.5 kg/ha. Detailed fertilization schedules and application methods are summarized in Appendix A.

### 4.2. Vectors Construction and Plant Transformation

For the genome editing of *OsERF34*, gene-specific target sequences were selected by the target-designing website (http://crispr.hzau.edu.cn/cgi-bin/CRISPR/CRISPR, accessed on 1 February 2022) and inserted into the Bsa I restriction site of the pMKO-Cas9-Os backbone. To construct Ubi: OsERF34-3FLAG, the coding sequences of *OsERF34* were amplified and cloned into Ubi-pPUN-3FLAG. Using mature embryos of rice (ZH11) as an explant material, rapidly and efficiently induce high-quality embryogenic callus tissues to serve as recipient materials for genetic transformation. Subsequently, infect the recipient tissues with Agrobacterium tumefaciens carrying recombinant binary vectors containing target genes, facilitating T-DNA insertion into the plant genome. Through rigorous antibiotic selection pressure corresponding to the vector’s resistance markers, screen and isolate independent resistant callus lines. These selected calli are then subjected to phytohormone-regulated differentiation protocols and plant regeneration processes, ultimately generating genetically transformed plant lines with a stable integration of exogenous DNA [56].

### 4.3. Rice Grain and Flour Sample Preparation

The mature rice grains were harvested in the field, dried naturally, and stored at room temperature for more than three months for appearance and processing quality measurement. The grain was first dehulled using a rice huller (Satake, Tokyo, Japan) and then polished using a ZM100 grain polisher (Xinfeng, Taizhou, China). The brown rice and polished rice were ground in a pulverizer (Cyclotec 1093, Foss, Sweden), respectively. The flour was sieved through a 0.5 mm screen. Brown rice flour was used for rice grain composition content analysis in order to avoid inconsistency in the grinding degree, leading to biased results. The milled rice flour was used for starch granules isolation, viscosity, pasting properties, and starch chain length distribution analysis.

### 4.4. Measurement of Appearance Quality

The grain length, width, and thickness of fully filled grains were measured using a sliding caliper with a precision of 0.001 mm. Ten randomly selected field-collected plump grains were measured in triplicate.

The degree of chalkiness of the milled rice was assessed visually and then used for the measurement of the grain chalkiness rate. The degree of chalkiness of the milled rice was assessed visually and then used for the measurement of grain chalkiness rate. The chalkiness degree of milled rice was visually evaluated to determine the chalky grain rate. One hundred randomly sampled grains were assessed, and the number of chalky grains (exhibiting white opaque regions in the endosperm, including ventral, central, or dorsal chalkiness) was recorded as the chalky grain rate. The chalkiness degree was quantified as the percentage of chalky area relative to the total surface area of the rice grain. Three replicates were carried out for each experiment.

### 4.5. Observation of Starch Granules in the Endosperm

Mature brown rice grains were transversely broken, and the ruptured transverse surface and milled rice powder (200-mesh) were coated with gold to prepare the samples. The starch granule morphology was observed and images were obtained using a scanning electron microscope (Hitachi S3400N, Hitachi, Tokyo, Japan). Starch granules were isolated according to a previous report [57]. The size distribution of the starch granules was determined by a laser-scattering particle size distribution analyzer (LA-960S2, Horiba, Tokyo, Japan). Three replicates were carried out for each experiment.

### 4.6. Physicochemical Properties Analysis

The total starch and amylose contents of mature endosperm flour were determined with the starch assay kits (Megazyme, Wicklow, Ireland). The protein and lipid contents in the grains were measured following the method described by Kang, et al. [34]. All of the composition content results were based on dry weight.

To determine the pasting properties, 3 g of milled rice powder (14% moisture basis) were transferred into a container with 25 mL of distilled water. The sample was mixed and measured with a Rapid Visco Analyzer (RVA Starch Master, Perten, Stockholm, Sweden) according to the manufacturer’s protocol. Three replicates were carried out for each experiment.

For the gelatinization temperature analysis, 5 mg of rice powder were placed in an aluminum sample cup, mixed with 10 μL of distilled water, and sealed. The samples were analyzed by a DSC following the manufacturer’s protocols [33].

To assay the starch structure, X-ray diffraction patterns of starch were obtained with a Bruker D8 analytical diffractometer according to Qin [57].

To determine the starch chain length distributions of amylopectin, 25 mg of milled rice flour was weighed and placed into a flat-bottom flask with a sealed lid. Then, 5 mL of NaAC buffer (0.05 M, pH = 3.5) were added. The mixture was heated at 130–140 °C and kept boiling for 30 min, after which it was cooled to 50 °C. Next, 25 μL of isoamylase (1000 U/L) were added, and the flask was incubated in a 40 °C constant-temperature shaker for 48 h to debranch the starch chains. The mixture was centrifuged at 12,000 rpm for 10 min, and 100 μL of the supernatant was dried using a centrifugal vacuum dryer. Maltose, a mobility marker, was added to the dried sample, and the mixture was dried thoroughly. Then, 3.5 μL of a sodium cyanoborohydride–tetrahydrofuran solution (1 M) and 3.5 μL of 8-aminopyrene-1,3,6-trisulfonic acid trisodium (APTS; 5 mg of APTS dissolved in 48 μL of 15% acetic acid) were added. The mixture was labeled for more than 4 h in the dark at room temperature. A 5 μL aliquot of the sample solution was transferred to a 200 μL centrifuge tube, and 195 μL of ultrapure water were added. The mixture was thoroughly combined for injection into a P/ACE MDQ capillary electrophoresis system (Beckman-Coulter, Fullerton, CA, USA) equipped with a laser-induced fluorescence detector and an argon lamp (light source). Separation was performed on a carbohydrate-coated capillary (I.D. = 50 μm, Beckman-Coulter), controlled and recorded by 32-Karat software (Version 7.0). Chain lengths (degrees of polymerization) were assigned based on the migration of maltose. The relative percentage of the degree of polymerization of the starch chains was calculated by determining the proportion of the integral area of each peak. Each experiment was conducted in triplicate. All data are presented on a dry-weight basis of rice flour.

### 4.7. Determination of Solubility and Swelling Capacity

Milled rice flour (400 mg dry basis, w_1_) was suspended in 12.5 mL ddH_2_O within a 20 mL stoppered graduated tube. The mixture was vortexed, equilibrated at 25 °C (5 min), incubated at 50 °C (30 min), cooled in an ice bath (1 min), and re-equilibrated at 25 °C (5 min). After centrifugation (3000× *g*, 15 min), the supernatant was oven-dried at 105 °C to a constant weight (w_2_). Starch solubility was calculated as (w_2_/w_1_) × 100%, while the swelling capacity was determined by the sediment mass relative to w_1_. Three replicates were carried out for each experiment. The data are calculated based on the dry weight of rice flour.

### 4.8. Processing Quality and Character Analysis

The brown rice rate, milled rice rate, and head rice rate were evaluated following the methods described for cooking rice variety quality from the agricultural industry standard of the People’s Republic of China NY/T 593-2021 [3]. The processing characteristics, including shearing hardness, shearing energy, shear fracture, compression hardness, compression energy, and compression fracture, of mature brown rice were tested using a rice texture analyzer (FTC, Sterling, VA, USA). In shear mode, the inductive probe is 1000 N, the initial force is 1.5 N, the deformation is 20%, and the shearing speed is 10 mm/min. In compression mode, the inductive probe is 100 N, the initial force is 0.5 N, the deformation is 20%, and the compression speed is 10 mm/min. Ten samples were randomly selected and analyzed in triplicate.

### 4.9. RNA Extraction and qRT-PCR

Total RNA was extracted using the Plant RNA Kit (Omega, Guangzhou, China) and reverse transcribed with a Hifair III First Strand cDNA Synthesis SuperMix for qPCR (Toyobo, Shanghai, China) according to the manufacturer’s instructions. Quantitative real-time PCR was performed in triplicate on a Light Cycler 480 real-time system with the Hieff qPCR SYBR Green Master Mix (Toyobo, Shanghai, China). The expression analysis was performed with three biologicals.

## 5. Conclusions

The AP2/ERF family is directly involved in various plant developmental and stress regulation processes. Here, we show that *OsERF34* is directly involved in rice grain quality, especially in processing and appearance quality. Mutants exhibiting higher chalkiness, with a looser starch arrangement, bigger granule size with less extrusion, higher relative crystallinity of starch, more short chains, and significantly reduced grain hardness, indicate that *OsERF34* affects processing and appearance quality mainly through starch granule synthesis, accumulation, and arrangement. Meanwhile, with the regulation of *OsERF34* on rice grain composition, all of these enable an increase in hardness of WT and OE grains, which could resist grinding fracture. Most importantly, the enhancement of grain hardness could improve the head rice rate without weakening other physicochemical properties. Future studies should focus on elucidating the precise molecular mechanism through which *OsERF34* regulates grain traits and investigating its interactions with other transcription factors (TFs) could further clarify the functional role of *OsERF34* in modulating grain quality. Additionally, exploring the functions of *OsERF34* among different rice varieties and environmental conditions may also help in developing rice lines with improved milling efficiency and grain quality.

## Figures and Tables

**Figure 1 plants-14-01633-f001:**
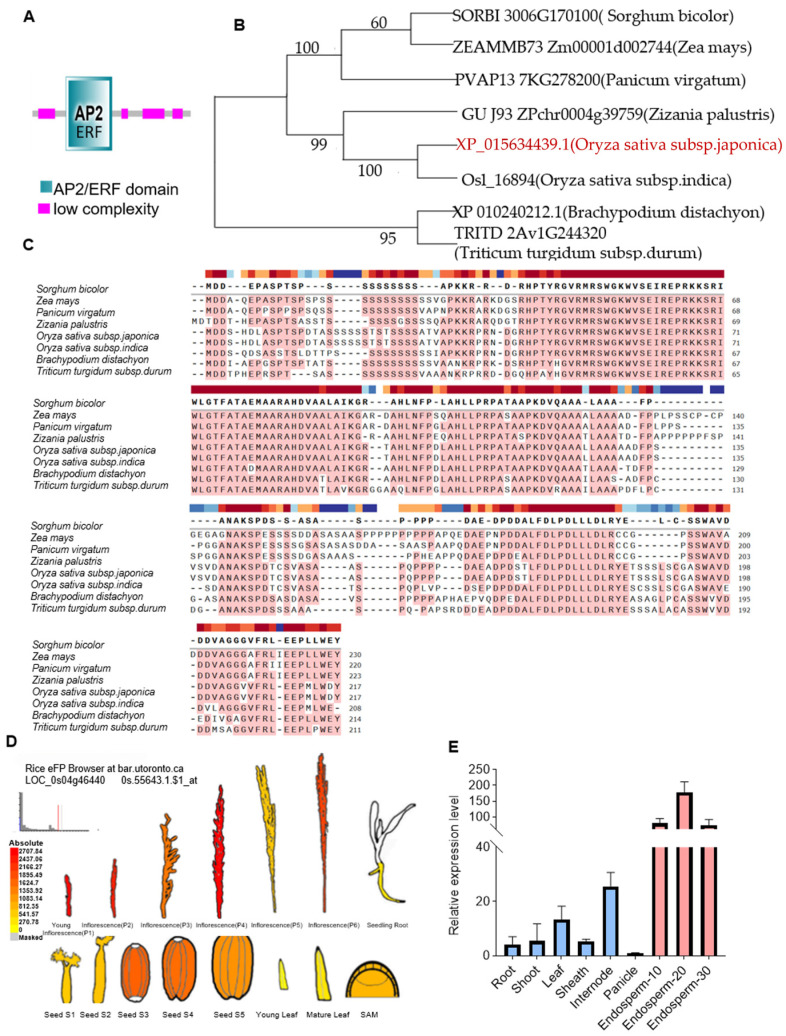
Bioinformatics expression pattern analysis of *OsERF34*. (**A**) The structure of AP2/ERF protein family gene *OsERF34*. (**B**) The phylogenetic tree analysis of OsERF34. Our target OsERF34 is highlighted in red. (**C**) Multiple sequence alignment. (**D**) Expression patterns of *OsERF34* based on the rice eFP browser. Seed stages S1–S5 correspond to *OsERF34* expression levels in the endosperm at 6, 12, 18, 24, and 30 days after fertilization (DAF). (**E**) Expression patterns of *OsERF34* in various tissues and developing endosperm at 10, 20, and 30 DAF in rice detected by quantitative real-time PCR (RT-qPCR). Error bars indicate SD (*n* = 3). The expression values were normalized to the *Ubiquitin* expression and shown relative to the minimum expression, which was set to 1.0.

**Figure 2 plants-14-01633-f002:**
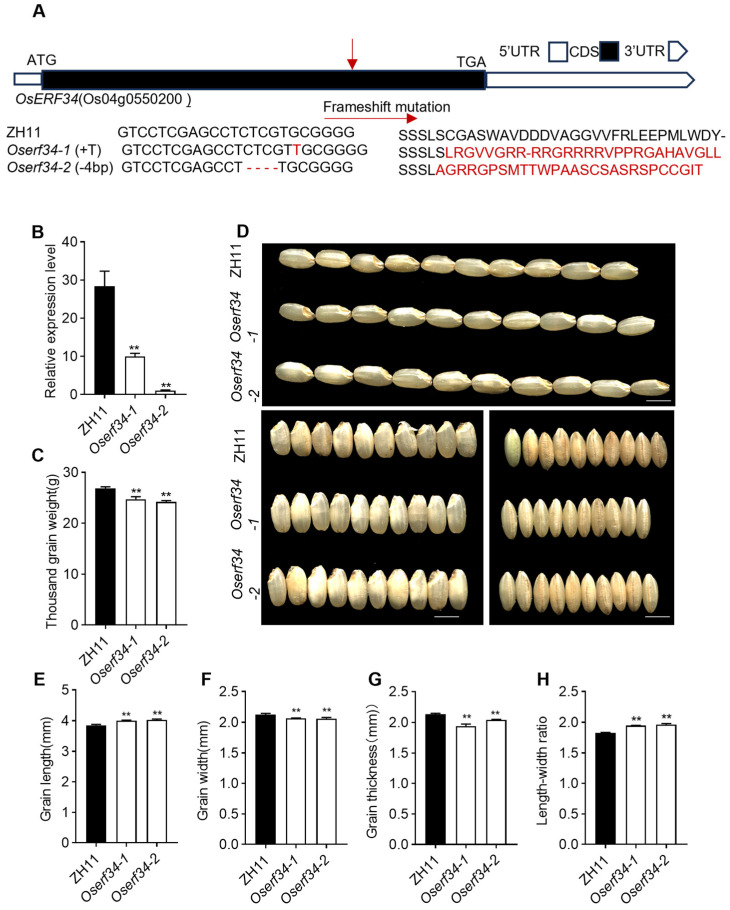
Analysis results of ZH11 and *Oserf34* on rice appearance. (**A**) the mutagenesis information of *OsERF34* generated by CRISPR-Cas9 in ZH11 genetic background. The schematic diagram shows the *OsERF34* gene with the CRISPR-Cas9 target sites indicated by arrows. Alignments between wild-type and mutated sequences containing the target sites (left) and amino acid frameshift mutation (right) are shown below. (**B**) Relative expression level of *OsERF34* in leaves of ZH11 and *Oserf34* by quantitative real-time PCR (RT-qPCR). Error bars indicate SD (*n* = 3). The expression values were normalized to the *Ubq* expression and are shown relative to the highest expression in each experiment, which was set to 100%. Asterisks indicate statistically notable differences (** *p* < 0.01, Student’s *t* test). (**C**) the thousand grain weight of ZH11 and *Oserf34*; (**D**–**H**) the phenotypic images ((**D**), scale line: 5 mm) and statistical charts of grain length (**E**), grain width (**F**), grain thickness (**G**) and length–width ratio (**H**) for ZH11 and *Oserf34*, respectively. (**E**–**G**) were measured using vernier calipers, with 10 randomly selected seeds per measurement and three replicates per sample. Error bars represent SD (*n* = 3); asterisks indicate statistically notable differences (** *p* < 0.01, Student’s *t* test).

**Figure 3 plants-14-01633-f003:**
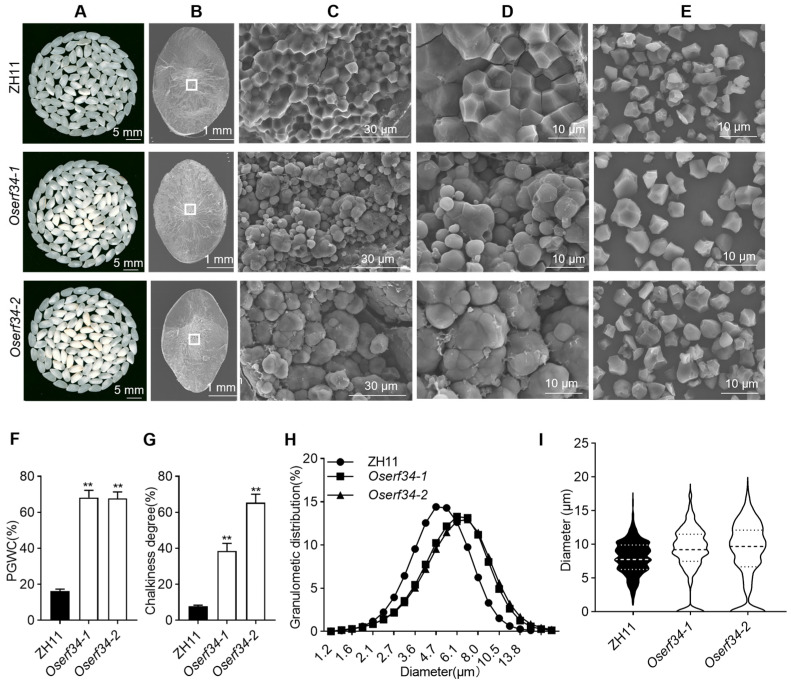
Comparison of chalkiness phenotype and starch granules between ZH11 and *Oserf34*. (**A**) Chalkiness images of ZH11 and *Oserf34*. (**B**–**D**) Scanning electron microscopy of cross-sections of endosperm in ZH11 and *Oserf34*, with scale bars of 1 mm (**B**), 30 µm (**C**), and 10 µm (**D**), respectively. The locations indicated by white rectangles correspond to the view positions of images (**C**–**E**), scanning electron microscopy of starch granules in ZH11 and *Oserf34*. (**F**,**G**), Percentage of Grains with Chalkiness (PGWC) and Chalkiness degree of ZH11 and *Oserf34*. Error bars represent SD (*n* = 3), asterisks indicate statistically notable differences (** *p* < 0.01, Student’s *t* test). (**H**,**I**), Distribution of starch granule sizes and diameters in ZH11 and *Oserf34*.

**Figure 4 plants-14-01633-f004:**
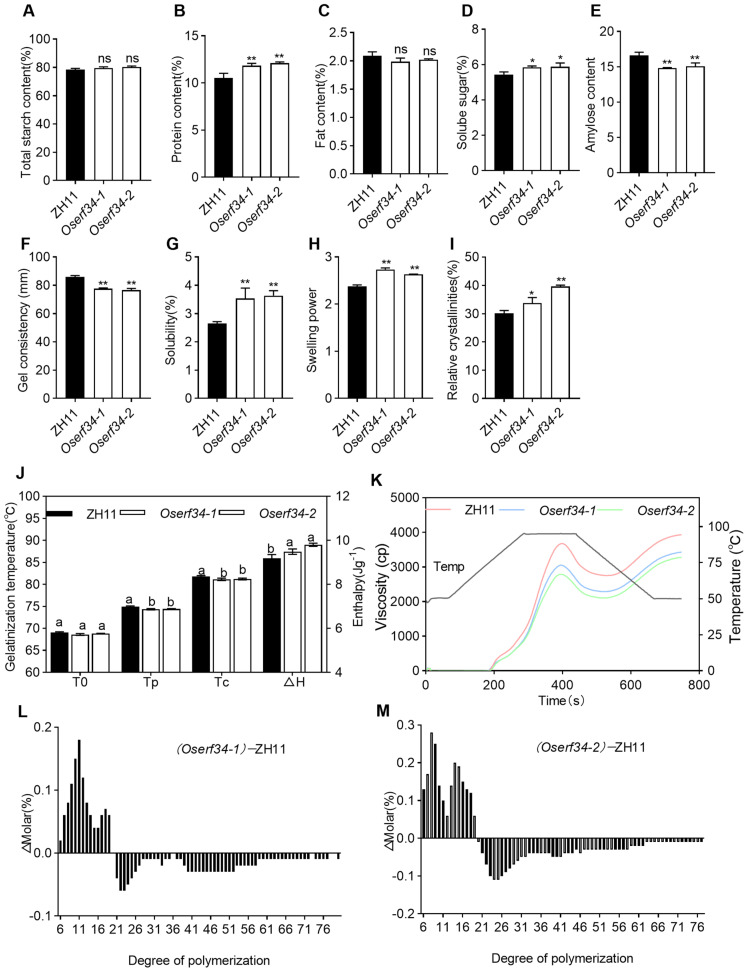
The physicochemical properties of the endosperm and starch granule size distribution in mature seeds of ZH11 and *Oserf34.* (**A**–**E**) Total starch content (**A**), protein content (**B**), fat content (**C**), Soluble sugar content (**D**), Amylose content (**E**) in brown rice flour of ZH11 and *Oserf34*; (**F**–**H**), gel consistency (**F**), water solubility (**G**), swelling capacity (**H**) of milled rice flour of ZH11 and *Oserf34*. (**I**) Relative crystallinity of starch in ZH11 and *Oserf34.* Error bars represent SD (*n* = 3). The asterisk and “ns” indicates a significant difference (* *p* < 0.05 and ** *p* < 0.01, Student’s *t* test) and no significant difference between *Oserf34* and ZH11, respectively. (**J**,**K**) differential scanning calorimetry (DSC) analysis (**J**) and rapid Visco analyzer (RVA) viscosity analysis (**K**) of milled rice flour of ZH11 and *Oserf34*. Different letters in (**J**) indicate significant differences (*p* < 0.05, one-way ANOVA). (**L**,**M**) Starch chain length distribution in ZH11 and *Oserf34* flour by mole percent of *Oserf34* flour minus mole percent of ZH11 flour [Δ (molar percent)].

**Figure 5 plants-14-01633-f005:**
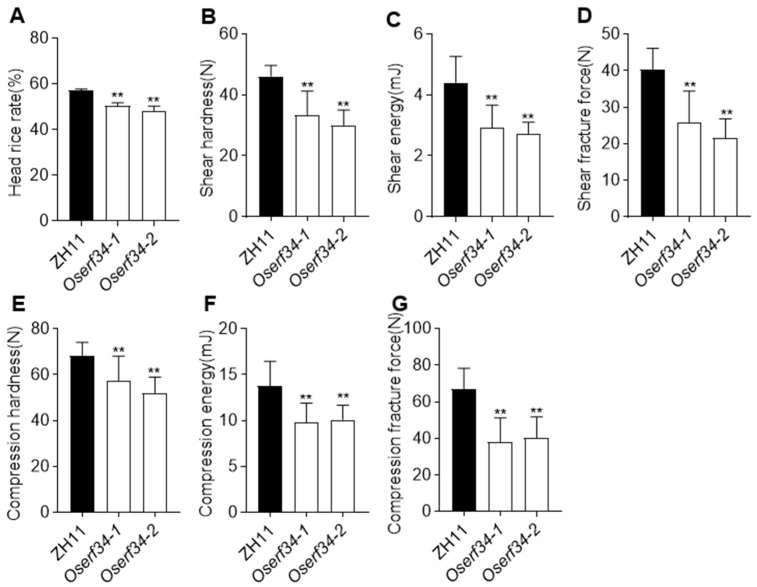
Rice processing quality and seed hardness analysis of ZH11 and *Oserf34.* (**A**) The head rice rate of ZH11 and *Oserf34*. Error bars represent SD (*n* = 3). The asterisk indicates a significant difference between *Oserf34* and ZH11 (Student’s *t*-test, *p* < 0.01). (**B**–**D**) Shear hardness (**B**), shear energy (**C**), and shear fracture force (**D**) of mature grains in ZH11 and *Oserf34*; (**E**–**G**) Compression hardness (**E**), compression energy (**F**), and compression fracture force (**G**) of mature grains in ZH11 and *Oserf34.* Error bars represent SD (*n* = 10); the asterisk indicates a significant difference between *Oserf34* and ZH11 (** *p* < 0.01, Student’s *t* test).

**Figure 6 plants-14-01633-f006:**
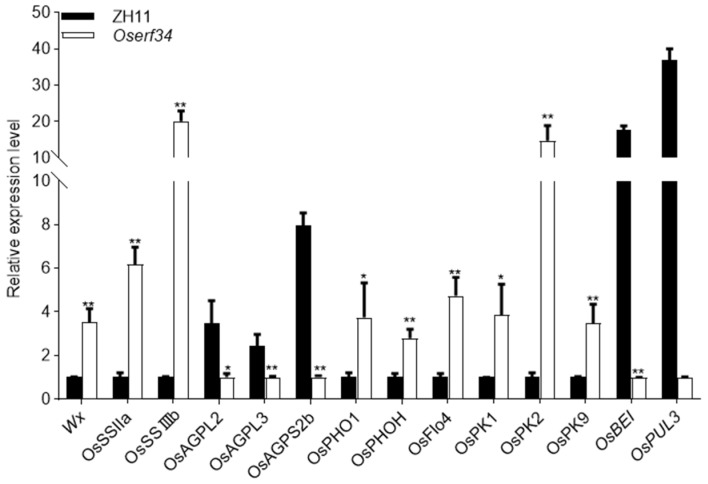
RT-qPCR analysis of related genes in ZH11 and *Oserf34.* RT-qPCR analysis of genes related to the synthesis of starch in ZH11 and *Oserf34.* Seeds at 20DAP of each genotype were collected for RT-qPCR analysis. The levels of each gene expression normalized to *Ubiquitin* expression are shown relative to the level in the lowest type, which was set to 1. Error bars denote SD (*n* = 3). Asterisks indicate significant differences between *Oserf34* and ZH11 (* *p* < 0.05 and ** *p* < 0.01, Student’s *t* test).

**Figure 7 plants-14-01633-f007:**
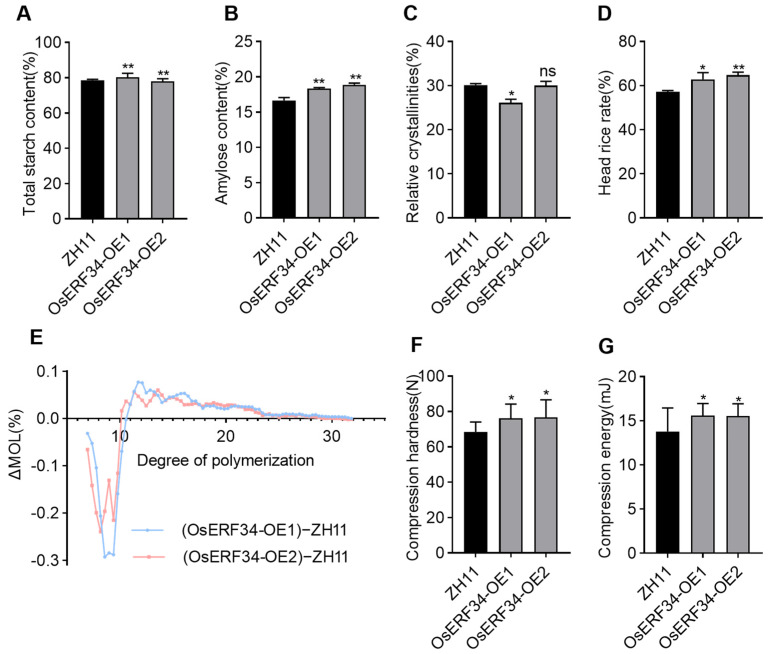
Comparative analysis of key phenotypic differences between ZH11 and OsERF34-OE. (**A**,**B**) Total starch content and amylose content in brown rice flour of ZH11 and OsERF34-OE; (**C**,**D**) the relative crystallinities of starch and head rice rate in ZH11 and OsERF34-OE. Error bars represent SD (*n* = 3), asterisks indicate statistically notable differences (* *p* < 0.05 and ** *p* < 0.01, Student’s *t* test), “ns” indicates no significant difference. (**E**) Starch chain length distribution in ZH11 and OsERF34-OE flour by mole percent of OsERF34-OE flour minus mole percent of ZH11 flour [Δ (molar percent)]. (**F**,**G**) Compression hardness (**F**) and compression energy (**G**) of ZH11 and OsERF34-OE. Error bars represent SD (*n* = 10), asterisks indicate statistically notable differences (* *p* < 0.05, Student’s *t* test).

## Data Availability

Date are contained within the article.

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
