# Peer review of "The Impact of OsERF34 on Rice Grain-Processing Traits and Appearance Quality"

_plants, 2025, doi:10.3390/plants14111633_

Round 1
Reviewer 1 Report
Comments and Suggestions for Authors
The manuscript submitted by Du et al., titled “The impact of OsERF34 on rice grain processing traits and appearance quality” explores the role of the transcription factor ERF34 in improving milling efficiency. Overall, the authors have demonstrated how ERF34 levels affect various factors affecting grain morphology and processing traits.
I do not have any major concerns regarding this manuscript; however, I do have several minor concerns. Overall, the manuscript needs to be checked for English and grammar as there are several grammatical errors that make it difficult to understand the message that the authors want to convey. In addition, the abstract also needs to be revised for an easier read.
There are perhaps too many figures in the manuscript. Some of the figures with the loss of function mutants and the OE lines can be consolidated such as figures 2 and 7, figures 3 and 9 and figures 4 and 8, as a lot of the same parameters are being determined and they can be compared side by side rather than going back and forth.
- The authors need to be consistent with labeling OsERF34. In the text, the species is always mentioned. For example, OsERF34 or Os-ERF34-OE or oserf34 whereas the figures and figure legends do not indicate “Os”. See the examples below for inconsistency:
- Line 92 - OSERF34 should be changed to OsERF34
- Line 103 – oserf34 and line 106 says Oserf34 – the labeling should be consistent
- Figure 1A legend - label it as structure of ERF protein family
- Figure 1D has a lot of text underneath that doesn’t fit in the figure and doesn’t quite make sense. The authors should put the citation for the eFP browser in the legend or in the text/references.
- Line 91: the citation for Figure 1A should be moved to line 89 where the authors talk about the gene structure of AP2/ERF TF.
- Lines 94-95: The authors mention that they found a close resemblance of OsERF34 to Zp04g39759 in Zizania but do not discuss the significance of this resemblance.
- Figure 1D-E – How do the seed stages in the eFP browser and the endosperm stages in the qRT-PCR compare?
- Lines 102-105 should be two separate sentences.
- Figure 2G: The differences don’t seem significant looking at the error bars but the authors have it listed as significant so it would help if the authors can provide data in a table format (could be supplemental).
- Figure 2: How many seeds were sampled per replicate? This detail is missing in the text/methods. What p-value indicates significance here? It is not mentioned in the legend.
- Figure 3B is not referenced in the text outside of the figure legend
- Lines 165-170 – Please revise for grammar/sentence structure
- Figure 4 legend is repetitive and should be simplified – for example: Total starch content (A), protein content (B), fat content (C) etc in ZH11 and erf34.
- Lines 186-187 – Specifically, Notably – revise
- Figure 4J – Do the letters a and b indicate significance? If so, it should be mentioned in the legend. The difference between * and ** should be clearly stated in the legend
- Supp figure 2B: y-axis label typo - milled rice rate %
- Figure 6: The legend says that there are asterisks indicating significance, however, there are none in the figure
- Figure 6: Is there a correlation between expression levels of all genes involved in starch synthesis or starch breakdown? The authors’ conclusions aren’t very clear. It would be great to have a supplemental figure or table indicating which process these genes are involved in.
- Lines 267-268 – “As observed in the mutants, OsERF34 strongly expressed in endosperm of rice seeds, we wonder grain morphology in the OsERF34-OE” – revise sentence for clarity
- Figure 7: the authors should clearly label the difference in p-value between * and ** in the legend
- Line 543-545 – There’s text from a template possibly?
- Table 2 is not referenced anywhere in the text
- Figure 8 – details about error bars and statistical significance are missing in the legend
The text needs to be edited by a native English speaker as there are several parts of the manuscript that were difficult to follow due to grammatical errors and incomplete sentences.
Reviewer 2 Report
Comments and Suggestions for Authors
Oserf should have the O from Oryza capitalized only the erf should be lowercase. Please fix in the entire document.
Please add to the intro the definitions of head rice rate and discuss what causes chalkiness lowering quality of the grain.
Line 69 Please reference your previous study in the text.
Why does the species not line up with the sequences listed in figure 1 C. Also why only one homoeologue is listed for the durum as durum is a tetraploid.
Please show in Figure 2 where you targeted ERF34 in part A. I think it is very late in the gene.
What tissue was used for the expression differences in figure 2B.
Please define PGWC in the text.
Sentence starting on line 165 does not make sense higher significant reduction? Greater reduction maybe.
Figure 6 states asterisks indicate significance, but no asterisks are in the figure. So is nothing significant, (hard to believe based on figure) or please add asterisks.
Why not condense the figures to show both the OE and KOs to better show the comparisons
Line 350 needs a citation(s).
Why is the expression of ERF34 lower in the mutants, I would be surprised if you managed to mutated the regulatory region using CRISPR that late in the gene. Please add to the discussion.
Please add more to your methods on growing conditions. Like light levels type of soil, fertilization levels, irrigation, field or glasshouse or chamber for transgenics etc.
Please give exact sequence of vectors used and CDS overexpressed in the suppl. texts.
In section 4.4 please give more detail from the previous report on what you did. It is fine to reference old lit but please ensure you include what your team actually did. Same for section 4.6, 4.8 and 4.9.
Reviewer 3 Report
Comments and Suggestions for Authors
Head rice yield is an important topic for study. Enhancing this trait is important for farmers' incomes.
Abstract
The following isn’t true. The work in this paper is a stepping stone towards precision breeding only.
“These insights enable precision breeding strategies to concurrently improve milling efficiency via OE, addressing critical bottlenecks in rice quality optimization.”
The Introduction/Literature Review is weak. The authors need to provide background for what is known about head rice yield and processing quality. This should be in terms of both genetic and environmental factors that influence these phenotypic traits.
Line 38
These are all quality measurements that were created decades ago. The authors that initially developed them should be shown respect by citing them. This is the same for all of the methods reported.
“Now there are six grading indexes including head rice rate, chalkiness degree, transparency, amylose content, gel consistency and alkali spreading value in the current cooking rice variety quality standard according to Cooking rice variety quality, Agricultural industry standard of the People’s Republic of China. NY/T 593-2021[3].”
A simple, clear Research Objective is needed just before the Results section.
Methods
The number of replications for each measurement are needed.
Is the data presented on an as-is moisture basis or a dry-weight basis?
How were samples chosen for every method? Rice with greater amounts of chalk will weigh less and thus not be distributed evenly throughout a container.
Details on the production of the research samples are needed. Where were the samples grown - in a growth chamber, greenhouse, or field? Specifics needed for example, are what amendments were used and amounts.
“All materials were grown at the experimental station of the China National Rice Research Institute in Hangzhou (Zhejiang Province, China, 30° 150 N).”
What method was used for – “All generated constructs were introduced into calli of ZH11 for rice transformation.”
Line 479
This doesn’t make sense. The method cited is for polydextrose, which is a synthetic polymer.
“To mitigate particle-size-induced matrix effects, brown rice fractions were designated for proximate composition analysis via NIRS (AOAC Method 2000.11)”
Determination of Polydextrose in Foods by AOAC Method 2000.11 chrome-extension://efaidnbmnnnibpcajpcglclefindmkaj/https://assets.thermofisher.com/TFS-Assets/CMD/Application-Notes/AN-147-IC-HPAE-PAD-AOAC-Method-2000-11-Polydextrose-Food-AN71606-EN.pdf
Line 481
Detailed methods and/or references are needed for the following:
“Contrastingly, milled rice derivatives underwent controlled micronization for multi-modal structural analyses encompassing starch granule ultra-structure (SEM-EDS), shear-dependent viscosity profiles (RVA 4500), thermal transition behavior (DSC 8000), and amylopectin chain-length polymorphisms (HPAEC-PAD). To ensure analytical precision, brown rice flour was employed for compositional analysis of 485 rice grains to prevent systematic bias arising from particle size heterogeneity. Conversely, milled rice flour was specifically processed for structural characterization including starch granule isolation, viscosity profiling, pasting properties, and amylopectin chain length distribution.”
Line 490
Detailed methods and/or references are needed for the following: ALL Measurements of appearance quality. For example, how was grain chalkiness defined? What percentage of a kernel must be chalky? How many kernels were examined? Method details needed for “The size distribution of starch granules was determined by a la-496 ser-scattering particle size distribution analyser (LA-960S2, Horiba, Japan). “ Simply stating the instrument used isn’t a method description.
Line 507
RVA company contact information needed.
Line 520
What is refined rice flour?
Line 526
The method details are needed.
“Starch chain length distributions were measured using a 527 PA800 plus pharmaceutical analysis system (carbohydrate labeling and analysis, Beck-528 man Coulter, USA, http://www.beckmancoulter. com/). “
Line 532
What is n NY/T 83–2017?
Results
Line 425
The authors can do a better job of explaining their results. For example, the following statement has been known for decades.
“These results indicate that starch synthesis is a complex biologcal process regulated by the coordinated action of many genes and hormones.”
Discussion
Line 344
Rice grain quality is a phenotypic trait not agronomic.
“Rice grain quality is an important agronomic trait and has been in the spotlight for a long time.”
Line 347
The following isn’t true. There are numerous studies published that have “explored the genetic factors influencing rice processing characteristics.
“However, relatively few studies have explored the genetic factors influencing rice processing characteristics and overall processing quality, high-lighting the need for further research on this area.”
Line 349
References are needed to support these statements of fact related to head rice yield and milling quality.
The Discussion is very weak in terms of the associations between starch content/structure and processing quality traits. There is nothing about the association of waxy kernels and head rice yield. The authors don’t seem to have a strong background in this area. Perhaps strengthening the Introduction/Literature Review will help them with this. If not, perhaps they can identify a rice quality expert to help them improve their paper. The authors need to discuss the limitations of their study. For example, the study is about head rice yield yet head rice yield wasn't measured. Why? What evidence is there that any of the measurements made in the study are predictive of head rice yield?
Comments on the Quality of English Language
Much of the paper is hard to understand due to poor English quality.
Author Response
"Please see the attachment.

Round 2
Reviewer 2 Report
Comments and Suggestions for Authors
Figure 1 description is not OK. This needs to be rewritten and read for typos.
lines 131 to 141 what homology are you referencing is it the amino acids or DNA sequence please add to the text?
Please add fertilization levels to section 4.1 not just a protocol. That is not reproducible.
Please add sequence information on the plasmid and you OsERF34 as a suppl. figure.
Your method of rice transformation needs a citation and far greater detail. This is fluff and people can not reproduce your methods.
are you sure you calipers can measure 1 micrometer?
